

# How representative are FLUXNET measurements of surface fluxes during temperature extremes?

Sophie V.J. van der Horst[1], Andrew J. Pitman[2], Martin G. De Kauwe[2], Anna Ukkola[3], Gab Abramowitz[2], Peter Isaac[4]

[1]Meteorology and Air Quality, Wageningen University, 6700 HB, Wageningen, the Netherlands.
[2]ARC Centre of Excellence for Climate Extremes and Climate Change Research Centre, University of New South Wales, Sydney, NSW, 2052, Australia
[3]ARC Centre of Excellence for Climate Extremes, and Research School of Earth Sciences, Australian National University, Canberra, ACT, 2601 Australia
[4]OzFlux Central Node, TERN Ecosystem Processes, Melbourne, VIC 3159, Australia

*Correspondence to*: Andrew J. Pitman (a.pitman@unsw.edu.au)

**Abstract.** In response to a warming climate, temperature extremes are changing in many regions of the world. Therefore, understanding how the fluxes of sensible heat, latent heat and net ecosystem exchange respond and contribute to these changes is important. We examined 216 sites from the open access Tier 1 FLUXNET2015 and Free-Fair-Use La Thuile datasets, focussing only on observed (non-gap filled) data periods. We examined the availability of sensible heat, latent heat and net ecosystem exchange observations coincident in time with measured temperature for all temperatures, and separately

for the upper and lower tail of the temperature distribution and expressed this availability as a measurement ratio. We showed that the measurement ratios for both sensible and latent heat fluxes are generally lower (0.79 and 0.73 respectively) than for temperature, and the measurement ratio of net ecosystem exchange measurements are appreciably lower (0.42). However, sites do exist with a high proportion of measured sensible and latent heat fluxes, mostly over the United States, Europe and Australia. Few sites have a high proportion of measured fluxes at the lower tail of the temperature distribution

over very cold regions (e.g. Alaska, Russia) and at the upper tail in many warm regions (e.g. Central America and the majority of the Mediterranean region), and many of the world's coldest and hottest regions are not represented in the freely available FLUXNET data at all (e.g. India, the Gulf States, Greenland and Antarctica). However, some sites do provide measured fluxes at extreme temperatures suggesting an opportunity for the FLUXNET community to share strategies to increase measurement availability at the tails of the temperature distribution. We also highlight a wide discrepancy between

the measurement ratios across FLUXNET sites that is not related to the actual temperature or rainfall regimes at the site, which we cannot explain. Our analysis provides guidance to help select eddy covariance sites for researchers interested in exploring responses to temperature extremes.



## 1 Introduction

Changes in the upper and lower tails of the temperature distribution are key characteristics of how global warming will impact climate (Hartmann et al. 2013). These expected changes in temperature are in line with a series of recent high-profile extremes witnessed across Europe (2003, 2010; Coumou and Rahmstorf, 2012; Schär et al. 2004; Barriopedro et al. 2011),

western North America (van Mantgem et al. 2009), the Amazon (2005, 2010; Philips et al. 2009; Lewis et al. 2011) and Australia (2012/2013; van Gorsel et al. 2018). Changes in temperature extremes are not only limited to the warm tail; the cold tail has also seen a notable change, with observed decreases in cold extremes particularly across North America (Wolter et al. 2015). Given the wide-ranging impacts of temperature on vegetation function (Berry and Björkman, 1980; Gunderson et al., 2009; Valladares et al., 2014; Van Gorsel et al., 2016; Kumarathunge et al., in press), health (McMichael and

Lindgren, 2011), socio-economics (McEvoy et al., 2012; Colombo et al., 1999; Zander et al., 2015) and land-atmosphere feedbacks (Fischer et al., 2007; Teuling et al., 2010; Miralles et al., 2012; Kala et al., 2016; Donat et al. 2017), projecting the impact of changes in temperature extremes is critical.

Our understanding of how temperature extremes will change is based on simulations using coupled climate models, e.g. the

Coupled Model Intercomparison Project (CMIP5) (Eyring et al., 2016). To build confidence in these model projections, models should be consistent with our understanding of changing temperature extremes, the impact on the vegetation and the associated feedback on the climate. However, current models are known to have key weaknesses in simulating both temperature extremes (Sillmann et al., 2013; Sippel et al., 2017) and the response of the vegetation to these extremes. For example, most climate models represent broad geographic regions with a single photosynthetic temperature response

function, which varies only with plant functional type (Smith and Dukes 2013; Lombardozzi et al. 2015; Mercado et al. 2018). This assumption would seemingly contradict empirical evidence showing that the temperature response of photosynthesis varies as a function of climate (Berry & Björkman, 1980; Gunderson *et al.*, 2009). Furthermore, studies show that plants adjust their temperature response of photosynthesis and respiration to changes in ambient temperature (Way and Sage 2008; Lombardozzi et al. 2015).


Improving how well models simulate temperature extremes, and how vegetation responds to these extremes requires empirical data. The global network of eddy-covariance towers (commonly known as FLUXNET), which includes over 900 sites and over 7000 site years (fluxdata.org, 2018) provides measurements of the exchange of carbon, energy and water between the land and the atmosphere. Therefore, eddy covariance measurements provide our best ecosystem-scale estimate

of the vegetation's response to heat extremes (Ciais et al. 2005; Teuling et al. 2010; Wolf et al. 2013; von Buttlar et al. 2018; Flach et al. 2018; De Kauwe et al. 2018) although some limitations inevitably remain (*e.g.* lack of energy closure, see Wilson et al, 2002). Although the length of the temporal records varies across sites, some sites extend back several decades allowing estimates of the impact of natural variability and climate trends on carbon, energy and water fluxes to be examined.

From each FLUXNET site, high-frequency (30 or 60 minute) measurements of the exchange of latent heat flux ($Q_{le}$), sensible heat flux ($Q_h$) and net ecosystem exchange (NEE) are available, alongside meteorological variables (including air temperature, net radiation, precipitation and relative humidity). The scale of recorded flux measurements ($\sim km^2$) is directly relevant for evaluating land surface schemes used in CMIP type climate models (e.g. Krinner et al., 2005; Abramowitz et al., 2008; Blyth et al., 2011). As a result, land surface modellers routinely use these data to parameterise and evaluate models for

extreme conditions. For example, the land surface is important in simulating land-atmosphere feedbacks as a landscape transitions into drought. Ukkola et al. (2016) used FLUXNET data to show systematic errors in how well models captured this transition of the vegetation during periods of water stress. In general, as the land surface dries, the surface energy balance tends to partition available energy increasingly towards $Q_h$ and less towards $Q_{le}$, which has important implications



for atmospheric temperature, moisture and atmospheric boundary layer depth (Seneviratne et al., 2010). This understanding of land-atmosphere processes was used by Miralles et al. (2014) to link soil desiccation to the amplification of extreme heatwaves via land surface feedbacks.

While eddy-covariance data have been widely used to examine the impact of temperature extremes, the measurement of temperature and the measurement of $Q_{le}$, $Q_h$ and NEE are independent. In this study, we focus on the relationship between measurements of temperature, and in particular extreme temperatures and concurrent measurements of $Q_{le}$, $Q_h$ and NEE. Our aim is to characterise, for example, whether direct observations of $Q_{le}$, $Q_h$ and NEE are biased towards the temperature mean and lacking at the tails of the temperature distribution, or whether they are biased to one tail of the distribution. If biases

exist, is this true for all FLUXNET sites, or are there specific regions or climates where the tails of the temperature distribution are rich with measurements of $Q_{le}$, $Q_h$ and NEE?

Here, we first investigate which parts of the temperature distribution have simultaneous measurements of $Q_{le}$, $Q_h$ and NEE for a given site. We then aggregate the answers to this question to ask which sites contain the most measured $Q_{le}$, $Q_h$ and

NEE relative to measured temperatures. This question is posed separately for the flux measurements over the whole temperature distribution and for the upper and lower tails of the distribution. We also seek to identify which FLUXNET site data are most suitable for analysing processes under extreme temperature conditions. In particular, we identify those sites with the most observations of $Q_{le}$, $Q_h$ and NEE since these would enable land surface model development and evaluation of elements of the surface energy, water and carbon budgets.

**2 Methods**

**2.1 FLUXNET data**

We use 165 site-based data sets from the FLUXNET2015 (November 2016 release; http://fluxnet.fluxdata.org/data/fluxnet2015-dataset/) and an additional 51 data sets from the FLUXNET La Thuile (http://fluxnet.fluxdata.org/data/la-thuile-dataset/) data release. Overall, our analysis is therefore based on 216 different site

data sets. A list of all sites used and associated information including vegetation type, location, the period of observations and references are provided in Supplementary Table 4. The data were pre-processed using the FluxnetLSM package (Ukkola et al., 2017). Variables LE_F_MDS, H_F_MDS and NEE_VUT_REF and TA_F_MDS were used from FLUXNET2015 for $Q_{le}$, $Q_h$, NEE and air temperature respectively and LE_f, H_f, NEE_f and Ta_f from La Thuile. These variables were accompanied by quality control (QC) flags to indicate whether the data were observed or gap-filled. These QC flags

facilitate the selection of data based on measurement quality. In this study, we focus only on the observed data, which is marked by the quality control flag 0 and exclude all other data.

Our goal is to identify those FLUXNET sites with data useful to explore land surface processes under extreme temperature conditions. To be representative a site requires a reasonable sample of measured data. We therefore first excluded any

FLUXNET and La Thuile sites with less than 8 months of observed data. We also excluded any sites with less than 50% of the temperature data having been measured (i.e. QC = 0) as distinct from gap filled or missing data (this excluded 14 sites). We also tested the sensitivity of our conclusions to data length. Given our focus on $Q_{le}$, $Q_h$ and NEE we excluded night-time data where incoming shortwave radiation was < 1 W m$^{-2}$. In case of erroneous measurements of shortwave radiation at night we excluded all data between 11 pm and 6 am. Thus, discussion of the availability of measured fluxes at the lower tail of the

temperature distribution focuses on daytime minimum temperatures.





## 2.2 Data processing

For each site, we first determine which time steps have measurements of temperature. If an observation of temperature is available (i.e. QC = 0) we explore if, for this same time step there are measurements of $Q_{le}$, $Q_h$ and NEE with a QC flag of 0. We then calculate the ratio of the number of measurements of each of the three fluxes relative to the number of temperature

measurements. For each site, this ratio was first calculated over the whole temperature distribution. Thus, per flux, the total number of measurements for $Q_{le}$, $Q_h$ and NEE were each divided by the total number of measured temperatures. In addition, this ratio was calculated for only the temperatures in the highest 2.275% of the temperature distribution, and separately for the lowest 2.275% of temperatures. These ranges approximate the data above and below two standard deviations from the mean. We did repeat our analysis using exactly the two standard deviations; this led to some qualitative differences in our

results because some sites lack enough measurements to provide reliable results where the temperature distribution was not normally distributed.

## 3 Results

Figure 1a shows the normalised frequency distribution of temperature, aggregated over all sites. Values range from about $-40^{o}$C to $40^{o}$C and are approximately normally distributed. However, the upper tail ends more abruptly than the lower tail.

Figure 1a also shows the normalised frequency of $Q_{le}$, $Q_h$ and NEE for different values of temperature. The shapes of the distributions for $Q_{le}$ and $Q_h$ are similar and measurements exist across the entire range of sampled temperatures. Not surprisingly, the normalised frequency of measurements for both $Q_{le}$ and $Q_h$ are lower than for measured temperature. Notably, the frequency of NEE is much lower than for $Q_{le}$ and $Q_h$. Figure 1b shows the ratio of the number of measurements of $Q_{le}$, $Q_h$ and NEE relative to the number of measurements of temperature. In all cases, the ratios increase as a function of

increasing temperature, indicating that fluxes are better sampled for warmer than colder temperatures. At the lowest temperatures, ratios $Q_{le}$, $Q_h$ and NEE range from ~0 to ~0.3 but these increase as temperatures increase to maximum ratios of ~0.8 at around $20^{o}$C for $Q_{le}$ and $Q_h$ and ~0.6 at around $30^{o}$C for NEE. A minor dip in ratios occurs at $0^{o}$C associated with the phase change of water, which most likely affects instrumentation. At the upper extreme of the temperature distribution ratios decline between $30^{o}$C and $40^{o}$C from ~0.8 to ~0.6 for $Q_{le}$ and $Q_h$ and from ~0.6 to ~0.5 for NEE. However, in each case a

secondary peak of high ratios occurs for the very highest temperatures. This peak is associated with temperatures > $44^{o}$C, which are rare and associated with measurements at Au-Cpr (68 individual measurements), AU-GWW (23), AU-Stp (24) and SN-Dhr (33). Of these, the Australian sites tend to have high measurement ratios and this peak at very high temperatures almost entirely reflects observations from Australian sites.

Figure 2 shows the geographic distribution of measurement ratios for $Q_{le}$ (Figure 6 provides the actual ratio values associated with each site and temperature range). The ratio over the whole temperature distribution shows most sites (63%) exceed 0.7 and some sites (5%) exceed 0.9 (Figure 2a). These ratios drop considerably if the lower tail (Figure 2b) is examined. Since the lower tail is calculated for each site independently this result is not surprising for mid- and high-latitude sites where snow, freezing and frosts would affect measurements. However, this result is more surprising in southern Europe and south-

eastern Australia where the lower tail is warm relative to some sites with higher ratios that are colder (e.g. Japan, northern China, Scandinavia). In contrast, for the upper tail, Figure 1c shows many (67) sites with ratios exceeding 0.9 (see also Figure 6). While we focus on the US, Europe and Australia, we note  sites in Japan, China, South America and Russia with ratios exceeding 0.9. We also note few sites with measurement ratios > 0.8 over some regions with very high temperatures, including Africa and the Middle East, and no sites  in e.g. India, Pakistan and Greece. Figure 3 shows a broadly similar result

for $Q_h$ although overall the ratios are higher (on average 0.79) than for $Q_{le}$ (on average 0.73). This is most apparent for the upper tail (Figure 3c) where many of the sites with ratios of 0.8-0.9 for $Q_{le}$ are above 0.9 for $Q_h$.





Figure 4 shows the geographic distribution of measurement ratios for NEE (see also Figure 8). There is a sharp contrast with the maps of $Q_{le}$ (Figure 2) and $Q_h$ (Figure 3) and the overall average is 0.42 compared to 0.79 for $Q_h$ and 0.73 for $Q_{le}$. In terms of the overall metric (Figure 4a), no sites exist with a ratio exceeding 0.9, only one exceeds 0.8 but 18 exceed 0.7. Two sites

exceed 0.7 for the lower tail (Figure 4b) located in the eastern US (US-Orv, US-Wi0). Multiple sites (11) over North America exceed 0.9 for NEE at the upper tail of temperatures (Figure 4c) together with isolated sites over Europe (IT-Tor, ES-Ln2), China (CN-HaM, CN-Cha, CN-Dan) and Australia (AU-Ade).

To examine these results further, Figure 5 shows the measurement ratios as a function of mean annual precipitation and

mean annual temperature. Since only a sub-set of temperature data and associated rainfall data meet the quality control criteria, the rainfall shown in Figure 5 cannot be compared with standard meteorological observations at these locations. For $Q_{le}$, $Q_h$, and NEE, Figure 5 shows the lack of high ratios for the lower tail relative to the upper tail and the low ratios for NEE compared to $Q_{le}$ and $Q_h$. At the upper tail, many sites (e.g. AU-Cpr, DE-Akm and US-NR1) exceed measurement ratios of > 0.9 for $Q_{le}$ and $Q_h$, but for NEE, most are in cool sites with low precipitation. In addition, Figure 5 shows little

relationship between temperature or rainfall and the measurement ratios. For example, some cool dry sites have high measurement ratios whereas others have low ratios. Similarly, some hot wet sites have high and some have low ratios for both $Q_{le}$ and $Q_h$ and for the upper tail of NEE. Few sites have high ratios for the overall temperature distribution or for the lower tail of NEE. In other words, the temperature or rainfall at a specific FLUXNET sites does not explain why some sites have a high frequency of flux measurements while other sites rarely observe $Q_{le}$, $Q_h$, and NEE. In figure 5, we observe 5-10

sites with high measurement ratios at temperatures above ~25°C for the upper tail and for $Q_{le}$, $Q_h$, and (to a lesser degree) for NEE; these are predominantly FLUXNET sites located over Australia.

We finally aggregate our analyses for the overall ratio, the lower tail and the upper tail separately for $Q_{le}$, $Q_h$ and NEE (Figures 6-8) and we identify each FLUXNET site in terms of the measurement ratio. Figures 6-8 are then combined in

Figure 9 to highlight those sites with high measurement ratios for all of $Q_{le}$, $Q_h$ and NEE and for just $Q_{le}$ and $Q_h$ for the overall metric (Figure 9a), the lower tail (Figure 9b) and the upper tail (Figure 9c). Taking the overall statistic first (Figure 9a, additional details are listed in Supplementary Table 1), no sites are found with $Q_{le}$, $Q_h$ and NEE ratios exceeding 0.9. Only two sites, both in the US (US-Whs, US-WiO) have measurement ratios above 0.8. If NEE is omitted, 19 sites are selected where both $Q_{le}$ and $Q_h$ ratios exceed 0.9 (Figure 9a, listed in Supplementary Table 1). These include eight sites over

the US, four sites over Australia, two over China, and a single site from Denmark, Germany, France, Italy, and Portugal. Even if the threshold is reduced to only $Q_{le}$ and $Q_h$ ratios exceeding 0.8, still no sites over South America, Africa and perhaps critically, for high temperatures Central America and the majority of the Mediterranean region. The freely available FLUXNET datasets provides no data over India, Pakistan or the Gulf States.

If we are interested in the lower tail of temperatures and we seek sites with measurement ratios exceeding 0.8 for each of $Q_{le}$, $Q_h$ and NEE, we have two choices (US-Orv, US-Wi0). If only $Q_{le}$ and $Q_h$ are needed, the choice widens to 18 sites with seven sites in Australia, four in the US, one in each of China, Canada and France (Figure 9b, Supplementary Table 2). Here, we note that very cold regions are poorly sampled with no sites in Alaska, Russia, the Himalaya, Greenland or Antarctica.

At the upper tail, 16 sites have ratios exceeding 0.9 for each of $Q_{le}$, $Q_h$ and NEE and are in Canada (7), the US (6), China (3), Spain (1), Australia (1) and Italy (1) (Figure 7c, Supplementary Table 3). If only $Q_{le}$ and $Q_h$ are required above 0.9 there are many sites (32) and above 0.8 there are three sites in South America, one in Botswana, several in the southern US and



southern Europe and one in Israel. No sites remain in India, Pakistan, the Gulf States, Central America and the majority of the Mediterranean region.

**4 Discussion**

The FLUXNET eddy covariance flux measurements are among the most valuable observations available for developing,
evaluating and benchmarking land surface models. Under future climate change, warming driven by radiative forcing is likely to amplify by changes in the partitioning of available energy between latent and sensible heat at the surface (e.g. Seneviratne et al., 2010; Miralles et al., 2014; Donat et al., 2018; Ukkola et al., 2018). This change in the partitioning, linked with soil desiccation or changes in stomatal conductance under higher $CO_2$ provides an amplification of the large-scale meteorology and can lead to more extreme conditions via the coupled land-boundary layer system (Seneviratne et al., 2010;
Miralles et al., 2014). As the continental surface warms, some regions will experience temperatures beyond the historical record. Building land models for CMIP-type climate models that properly capture mechanisms and processes occurring in a region experiencing higher temperatures is helped if observations from other regions already experiencing those temperatures are available (so called climate analogues, or space for time substitutions). In this context, observations from FLUXNET are particularly valuable if they sample existing hot locations, and if they actually measure fluxes at those
locations at the upper tail of temperature.

Our results highlight multiple positives for those wishing to probe vegetation responses to temperate extremes and/or evaluate land surface models. Figure 9 shows many sites with high measurement ratios for $Q_{le}$ and $Q_h$ at the upper and lower tail, indicating a rich source of available observations. Conversely, if we seek observations of $Q_{le}$, $Q_h$ and NEE, these data are
more limited with only two sites with a measurement ratio > 0.8, and none > 0.9 at the lower tail and 16 sites at the upper tail (see Supplementary Tables 2 and 3). Of course, the > 0.9 measurement ratio is arbitrary and more sites become available at lower ratios; however, it is somewhat confronting that at > 0.8, 87% of the sites in Supplementary Table 2 are located in Europe, North America and Australia and for the upper tail, 88% of the sites in Supplementary Table 3 are located in these three regions. The non-Europe, North America and Australia sites are not distributed globally: Figure 9 shows virtually no
sites with high (> 0.8) measurement ratios in the tropics, Africa or South America for $Q_{le}$ and $Q_h$, and no sites at all in India or the Gulf states. These typically hot regions may be surrogates for how continental surfaces behave under future climate scenarios in the mid-latitudes and it is unfortunate that FLUXNET lacks observations in these regions.

In the absence of measurements from hot regions, the availability of observations from Australia becomes particularly
important because these sites cover a wide rainfall gradient, ranging from water through to energy limited sites. We note two possible reasons for the lack of freely available data in many regions. First, there may be a lack of sites, or sites that exist, may have low measurement ratios. Second, the high number of sites identified in our analysis with high measurement ratios located in Europe, North America and Australia largely reflects the high number of sites in the FLUXNET data. Similarly, the low number of sites in Africa, South America, India and the Gulf States reflects the rarity of FLUXNET sites in these
regions. There are, however, four sites in Africa, three in South America and one in Israel in FLUXNET, but these are excluded due to the shortness of the data record, and the low temperature measurement ratios. This is not intended as a criticism; it is a consequence of history (where groups grew with the capacity to maintain measurements and the common desire to run measurement sites near home institutions).

One result from our analysis is that overall, measurement ratios for $Q_h$ are higher than for $Q_{le}$ and both of these are much higher than NEE. This is true for the overall distribution of temperatures, and for the lower and upper tails of the distribution.





This result can be quickly visualised by comparing Figures 6, 7 and 8. In part, this is associated with the actual temperatures at the sites influencing the measurement ratios once aggregated. Figure 5 shows that the measurement ratios are generally lower at the lower tail than the higher tail for $Q_{le}$ and $Q_h$. Furthermore, for the lower tail, the ratios are generally lower at colder temperatures than warmer temperatures. We propose multiple reasons explaining these findings. First, $Q_h$, $Q_{le}$ and

NEE are all products of turbulent transport. While there have been significant improvements in instrumentation over the last 20 years, measurements of these fluxes over long periods and across a range of weather conditions remains challenging.

Measurement ratios of $< 1$ for $Q_h$, $Q_{le}$ and NEE are expected due to data loss caused by instrument failure, precipitation, ambient conditions that violate the assumptions of the eddy covariance method (particularly low- or non-stationary

turbulence) and other artefacts (Foken et al 2010; Burba 2013). The lower ratios for $Q_{le}$ in comparison to $Q_h$ are likely to be associated with measurement methods. The majority of sites use a sonic anemometer and an open-path gas analyser to measure $Q_h$, $Q_{le}$ and NEE. Both devices use measurement techniques over a physical path (sound waves for the sonic and infra-red for the open-path gas analyser). Anything that partially obscures the measurement path (condensation, mist, drizzle, snow, ice, etc) can interfere with the measurements. The sonic anemometers are robust to all but very intense rain but the

open-path gas analysers are more sensitive to anything that blocks the optical path (Foken et al., 2010). The $Q_h$ measurements only involve the sonic anemometer while $Q_{le}$ and NEE use measurements from the sonic (for vertical velocity component) and from the open-path gas analyser (for water and $CO_2$ concentration). Measurements for $Q_{le}$ and NEE are therefore inherently more complex than for $Q_h$, which explains the lower measurement ratio for $Q_{le}$ relative to $Q_h$.

The lower ratios at lower temperatures are likely to be associated with the occurrence of condensation (dew), which is more common at cooler temperatures; hence the observed dependence of the ratio on measured air temperature. However, the assumptions underpinning the measurement of surface fluxes using the eddy covariance method are violated in low turbulence conditions, which occurs mostly at night (excluded in our analysis) and low temperatures (e.g. at dawn where radiative cooling leads to a stable surface layer). For fluxes that are significantly different from 0 at night (e.g. NEE due to

ecosystem respiration) this leads to an overwhelming bias in the measurements unless low turbulence conditions, where the assumptions of the eddy covariance method fail, are excluded from the analysis. Therefore, friction velocity (u*) is used as a proxy for turbulence, by finding the site-specific value for u* above which NEE is independent of u* and removing all observations when u* is below this threshold (Aubinet et al., 2012). This often results in less than 20% of NEE data being available for estimating ecosystem respiration. The application of this turbulence filter causes the ratio for NEE to be much

lower than the ratio for $Q_h$ and $Q_{le}$. The occurrence of these conditions is more likely in lower temperature conditions, contributing to the slope in Figure 1b.

Our analysis has a specific weakness, which requires consideration when interpreting our results. There may be a temptation to interpret the ratios we report as a metric linked with measurement quality. To discourage such a temptation we draw

attention to two hypothetical FLUXNET sites, one with ratios around 0.9 and another around 0.3. In the former, the efforts around measurement quality are superficial and data are included unless a specific problem identified. At the latter, the efforts around measurement quality are rigorous and any doubts whatsoever about the data leads to it being discarded. For the latter case, one would suggest that the resulting data reported to the FLUXNET2015 or La Thuile archive are likely of the highest quality and most reliable to use in process-level examination of models or understanding of the surface energy

and carbon balance. The more complete data in the former example could in fact be misleading. In short, our analysis does not report on data quality, it only relates to coincident data availability and identifies those sites where measurements are available with high frequency and with a QC = 0.





Our methodology contained several assumptions, for example we excluded sites with less than 8 months of data. We tested the sensitivity to this assumption, examining whether the sites identified with high measurement ratios changed if we required 12 months of data. If we set a minimum length of record as 12 months, US-Wi0 (one of two sites with $Q_{le}$, $Q_h$ and NEE > 0.8), US-SP1, US-Orv and ES-Ln2 are excluded in Supplementary Table 1. The only sites with $Q_{le}$, $Q_h$ and NEE >

0.8 are excluded from the lower tail (US-Orv and US-Wi0), along with DK-Fou, US-SP1, and NL-Lan. At the upper tail multiple sites are excluded (AU-Rob, PT-Mi1, NL-Lan, Es-Ln2, US-Wi0, US-SP1 and US-Bar) are excluded. Therefore, requiring a 12-month data set has a significant impact on some of the otherwise most useful sites. Given the purpose of our analysis is to examine the tails of the distributions, we suggest that imposing longer measurement periods than absolutely required may prove counterproductive. In addition, we examined two other attributes of the FLUXNET data – whether our

measurement ratio changes between the first half of the data and the second half (i.e. to examine whether the measurement ratio improved over time) and whether any relationship exists between the total number of QC=0 observations and the measurement ratio. The first analysis found no evidence that higher measurement ratios were apparent in the first or second half of the data, something that might have been expected if the ability to sustain measurements improved over time. The second analysis also found no evidence of a relationship between the measurement ratio and the length of data

(Supplementary Figure 1).

One obvious criticism of our measurement ratio metric is the temptation to interpret the results as a way to select FLUXNET sites for model development and evaluation without further thought. Clearly, a high measurement ratio is only one aspect of a valuable data set. A modeller might, for example, prefer a large number of actual measurements with a low overall

measurement ratio rather than a site with few measurements but a high overall measurement ratio. We have noted above that we find no correlation between data length and measurement ratio but some sites (see Supplementary Tables 1-3) have both high measurement ratios and large amounts of data and others have high measurement ratios and low amounts of data. For example, the two sites with the highest measurement ratios overall (US-Whs and US-Wi0) sharply contrast on the amount of data (63,619 and 4621 temperature measurements respectively). In this case, US-Whs covers 2922 days of measurement and

93% of the time temperature data are reported (Supplementary Table 1), whereas US-Wh0 only measures for 365 days and only 62% of the time temperature data are reported. In contrast, sites such as CA-NS1 and CA-NS3 display very similar measurement ratios for $Q_{le}$, $Q_h$ and NEE, both cover 1826 days but CA-NS1 includes 30,269 temperature measurements while CA-NS3 includes only 22,689 temperature measurements. Clearly, many characteristics of a data set make it valuable for model development or model evaluation and our analysis should be viewed only as one of these characteristics. One way

forward to resolve how to choose FLUXNET data for extremes is to combine an analysis of meteorological sites with FLUXNET sites. Using sites maintained by meteorological agencies to identify extreme events (e.g. heatwaves) and then interrogate the FLUXNET sites near to the meteorological site for the availability of measurements of $Q_{le}$, $Q_h$ and NEE could enable a modeller to choose suitable sites for land surface model development and evaluation.

We also examined whether the measurement ratio varied by time of day for each site (Supplementary Figure 2). These examples are provided to illustrate individual site behaviour and to emphasise that major variations at each site are present. At Au-ASM, a weak diurnal cycle is visible in the measurement ratio with very similar and consistently high ratios of $Q_{le}$, $Q_h$ and NEE being slightly lower. At a second Australian site, AU-Tum measurement ratios increase from dawn through the day, and then drop off just before dusk. At CA-NS4 behaviour is similar to AU-ASM until late in the day when the

measurement ratios drop sharply. At DE-Hai there is little variation though the day and $Q_h$ is much higher than $Q_{le}$, and only NEE shows any diurnal variation. DE-Meh shows $Q_{le}$ and $Q_h$ are consistent through the day and are almost identical. DK-NuF shows $Q_{le}$ and $Q_h$ falling from dawn to around 10am, then stabilising at low value (~0.3-0.4) and then increasing strongly from 2pm to ratios > 0.7 while NEE increases weakly from ~0.2 gradually though the day. It-Tor shows little



diurnal variation in $Q_{le}$ and $Q_h$, but there is a strong diurnal variation in NEE. Finally, US-Whs shows high measurement ratios for $Q_h$ and $Q_{le}$, but falling slightly through the day with NEE increasing strongly from dawn to 11am, and then slowly declining through the day. If we assume that the hottest part of the day is around 1pm, those sites that provide useful observations of $Q_h$ and $Q_{le}$ coincident with these temperatures clearly require site-by-site evaluation. Thus, if sites are being
composited, the knowledge that different sites sample different parts of the diurnal cycle, and sample $Q_{le}$, $Q_h$ and NEE differently across the diurnal cycle needs to be taken into account.

Our analysis poses interesting questions about the FLUXNET data that deserve further exploration. Why do sites with a similar climate vary so greatly in terms of their frequency of reporting of $Q_{le}$, $Q_h$ and NEE in comparison to temperature?
Why are some sites able to do this routinely while others cannot, and can expertise be shared to resolve this? What are the implications of aggregating FLUXNET data given the large variations in which parts of the temperature distribution are sampled? Why are there major variations in the measurement ratios between sites over the diurnal cycle and what does this mean in terms of using site data from FLUXNET? Clearly, the FLUXNET data do provide our best ecosystem-scale estimate of the vegetation's response to heat extremes (Ciais et al. 2005; Teuling et al. 2010; Wolf et al. 2013; von Buttlar et al. 2018;
Flach et al. 2018; De Kauwe et al. 2018) but given the need to build land models representing extreme conditions these data cannot be used without further evaluation of the specific site data. We do not know if there are opportunities for the global community to prioritise new sites in regions that currently lack data, or directly support those measurements in regions with low measurement ratios. However, we suggest value in either new sites, or investment in existing sites, in countries that experience temperatures that are higher than those experienced across North America and Europe to enable land models to
be developed in anticipation of further warming.

## 5. Conclusions

We have examined the FLUXNET data by evaluating the availability of $Q_{le}$, $Q_h$ and NEE observations at time steps where temperature is measured (with a quality control flag QC = 0). We have analysed this spatially to identify those sites with a high availability of flux measurements, relative to temperature measurements, across the whole temperature distribution, and
at the upper and lower tails of the distribution.

Virtually all sites (~90%) with high measurement metrics for $Q_{le}$, $Q_h$ and NEE, or just $Q_{le}$ and $Q_h$, whether examining the whole distribution or just the lower tail or just the upper tail, are located in North America, Western Europe and Australia. There are no sites in India, South America, Africa, the Middle East and few sites in China. In terms of vulnerability, the
freely available FLUXNET data therefore cover regions representing 12-14% of the global population. Indeed, the poorest country with measurements (based on Gross Domestic Product, Portugal) suggests all countries ranked from Portugal (ranked 47[th]) to the poorest country (ranked 211[th]) lack any measurements. Another perspective is if countries are ranked on average temperature, none of the warmest 98 countries contain a site and Australia is the hottest country with sites with high measurement ratios. Conversely, North America, Western Europe and Australia have multiple sites with observations of $Q_{le}$
and $Q_h$ and some with NEE with high measurement ratios for both the lower and upper tail of the temperature distribution. For these three regions therefore, FLUXNET data provide a rich source of data for understanding how fluxes of energy, water and carbon behave under extreme temperature conditions. Overall, we have noted more frequent observations of $Q_h$ than $Q_{le}$ and both these fluxes are much more common than NEE. This leads to two conclusions: first, some regions, particularly very hot regions that will be first to experience novel climates require observations. Less obviously, we highlight
a wide discrepancy between the measurement ratios across FLUXNET sites that is not related to the actual temperature or rainfall at the site. Clearly, some sites seem able to retrieve $Q_{le}$, $Q_h$ and NEE reliably at extreme temperatures while others



cannot. This may provide an opportunity for the FLUXNET community to share best practice strategies to identify ways to ensure measurements at the tails of the temperature distribution.

Finally, we restate a key caveat to our paper to avoid any misunderstanding. Our analysis does not highlight the "best data".
A site might have high ratios because of poor QC control, or low metrics because of strict controls. However, our paper does highlight sites with frequent observations of $Q_{le}$, $Q_h$ and NEE coincident with temperature observations where all have a QC = 0. A modeller might of course reject some of these sites for reasons of data record length, vegetation type, soil type or a multitude of other reasons. However, we suggest that our analysis provides one way for modellers to identify sites from the FLUXNET archive that warrant closer scrutiny for development and evaluation of land surface models under extreme
temperature conditions.

*Author contributions.* The ideas for this study originated in discussions with all authors. SH carried out the analysis supported by all authors. The manuscript was prepared with contributions from all authors.

*Code availability.* All code is freely available from*:* https://github.com/sophievanderhorst/FLUXNET

*Data availability.* All eddy covariance data are available from http://fluxnet.fluxdata.org/data/fluxnet2015-dataset/ and http://fluxnet.fluxdata.org/data/la-thuile-dataset/.

*Competing interests.* The authors declare no competing financial interests.

*Acknowledgements.* AJP, MDK, AU and GA acknowledge support from the Australian Research Council Centre of Excellence for Climate Extremes (CE170100023). SH would like to thank Prof. Bert Holtslag of Wageningen University for his comments on the manuscript and his help in arranging the internship. This work used eddy covariance data acquired by
the FLUXNET community and in particular by the following networks: AmeriFlux (U.S. Department of Energy, Biological and Environmental Research, Terrestrial Carbon Program (DE–FG02–04ER63917 and DE–FG02– 04ER63911)), AfriFlux, AsiaFlux, CarboAfrica, CarboEuropeIP, CarboItaly, CarboMont, ChinaFlux, Fluxnet–Canada (supported by CFCAS, NSERC, BIOCAP, Environment Canada, and NRCan), GreenGrass, KoFlux, LBA, NECC, OzFlux, TCOS–Siberia, USCCC. We acknowledge the financial support to the eddy covariance data harmonization provided by CarboEuropeIP,
FAO–GTOS–TCO, iLEAPS, Max Planck Institute for Biogeochemistry, National Science Foundation, University of Tuscia, Université Laval and Environment Canada and US Department of Energy and the database development and technical support from Berkeley Water Center, Lawrence Berkeley National Laboratory, Microsoft Research eScience, Oak Ridge National Laboratory, University of California, University of Virginia.






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





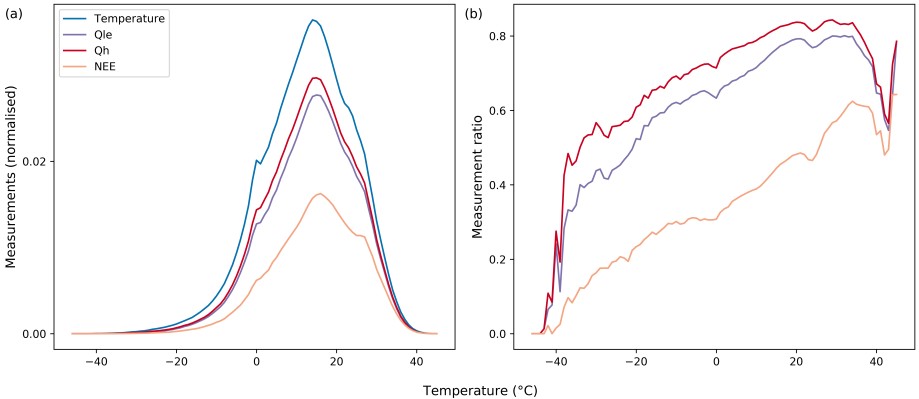

Figure 1: Availability of temperature, $Q_{le}$, $Q_h$, and NEE measurements in each 1 °C temperature bin. Panel (a) shows the normalised number of measurements of temperature, $Q_{le}$, $Q_h$ and NEE. Panel (b) shows the ratio of $Q_{le}$, $Q_h$ and NEE measurements relative to temperature measurements.



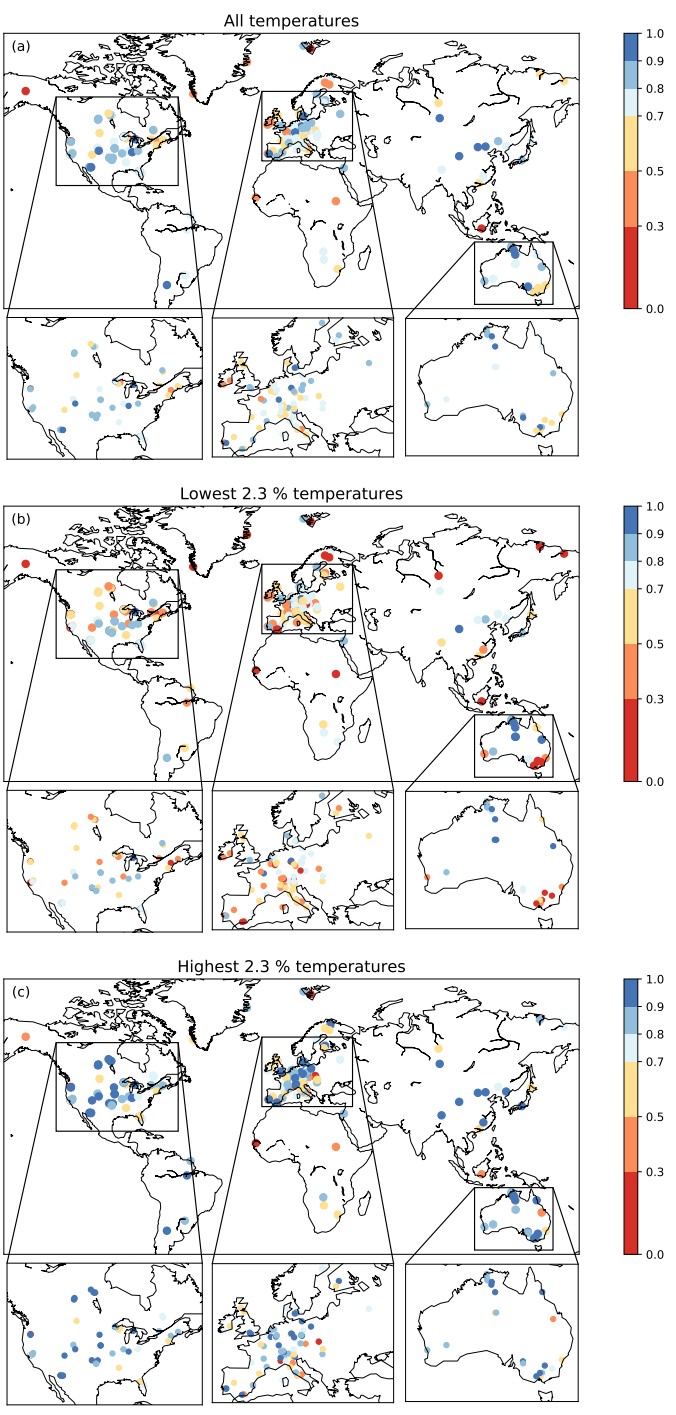

Figure 2: Maps of $Q_{le}$ measurement ratios. Panel (a) shows the $Q_{le}$ measurement ratios for the overall temperature distribution, panel (b) shows for the lower extreme and panel (c) for the upper extreme. Each dot on the map represents a flux tower site.




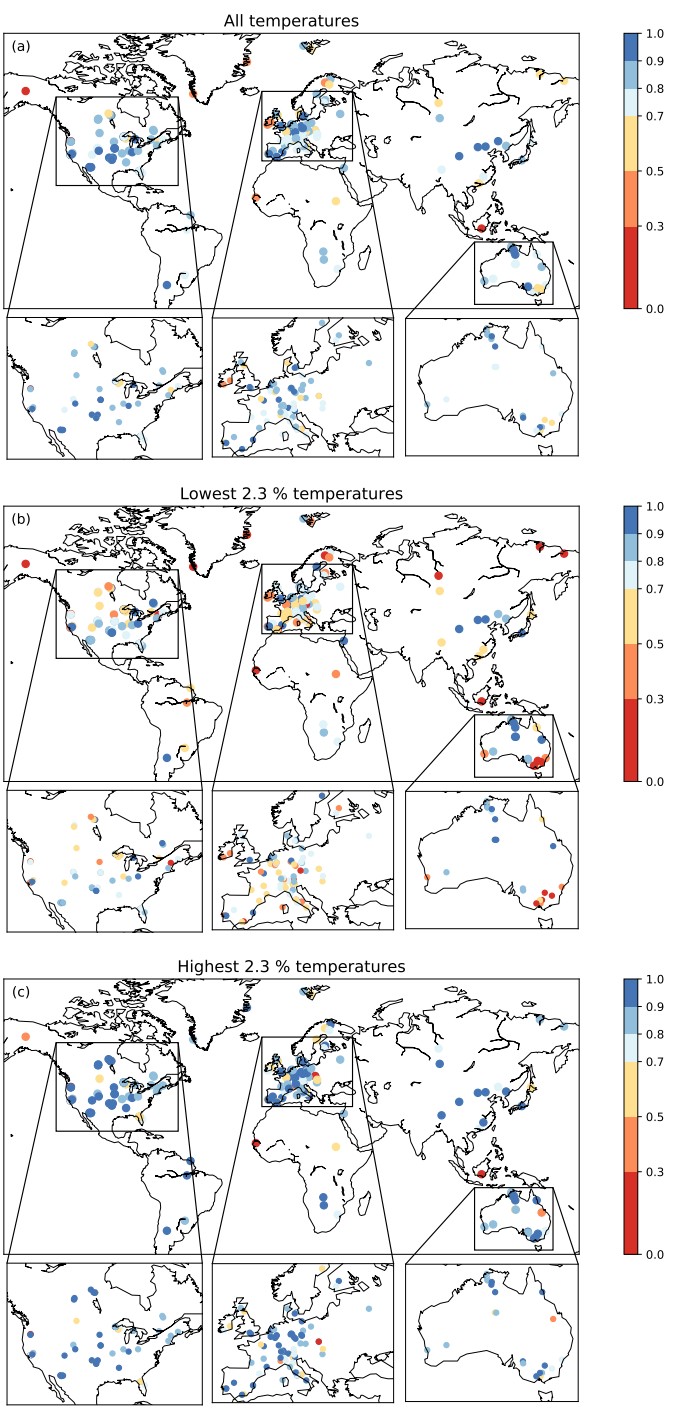

Figure 3: Maps of $Q_h$ measurement ratios. Panel (a) shows the $Q_h$ measurement ratios for the overall temperature distribution, panel (b) for the lower extreme and panel (c) for the upper extreme. Each dot on the map represents a flux tower site.



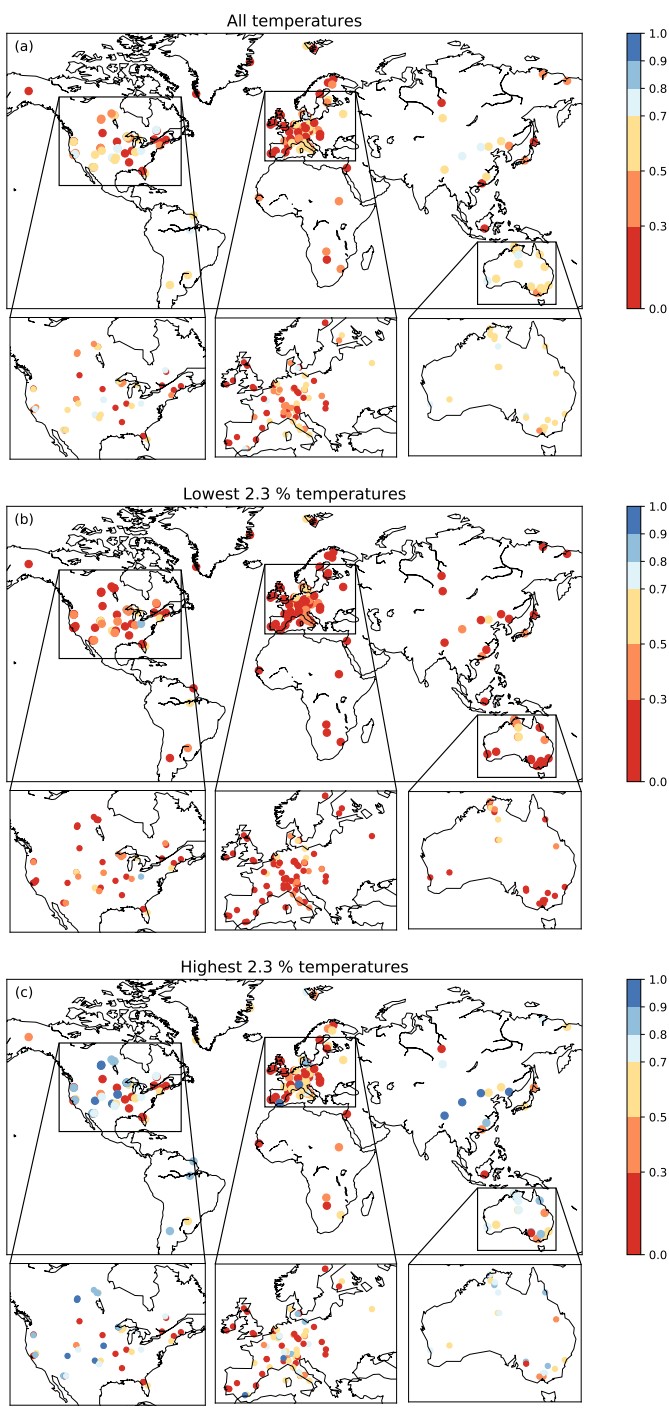

Figure 4: Maps of NEE measurement ratios. Panel (a) shows the NEE measurement ratios for the overall temperature distribution, panel (b) for the lower extreme and panel (c) for the upper extreme. Each dot on the map represents a flux tower site.



Figure 5: Measurement ratios as a function of mean annual temperature and precipitation. Panel (a) shows the measurement ratios for the overall temperature distribution, the lower extreme and upper extreme for $Q_{le}$, panel (b) for $Q_h$ and panel (c) for NEE, respectively.



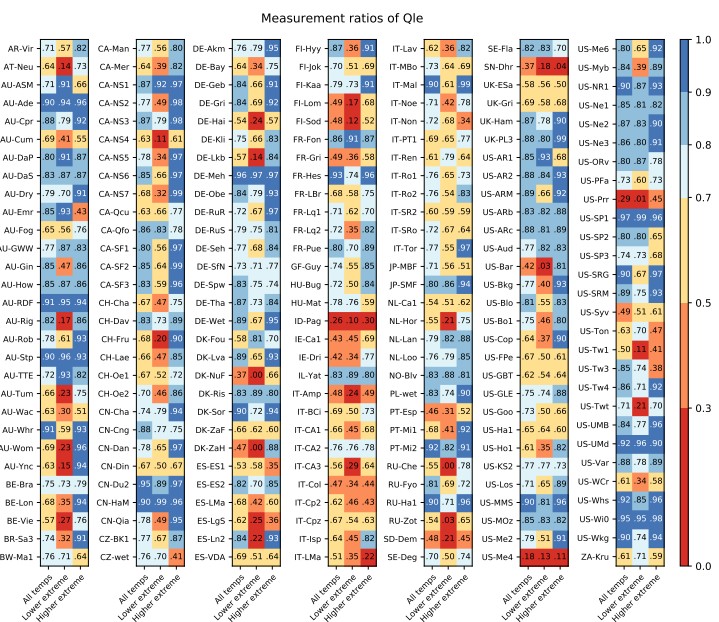

Figure 6: $Q_{le}$ measurement ratios of flux tower sites for all temperatures, the lower extreme temperatures, and the upper extreme temperatures.

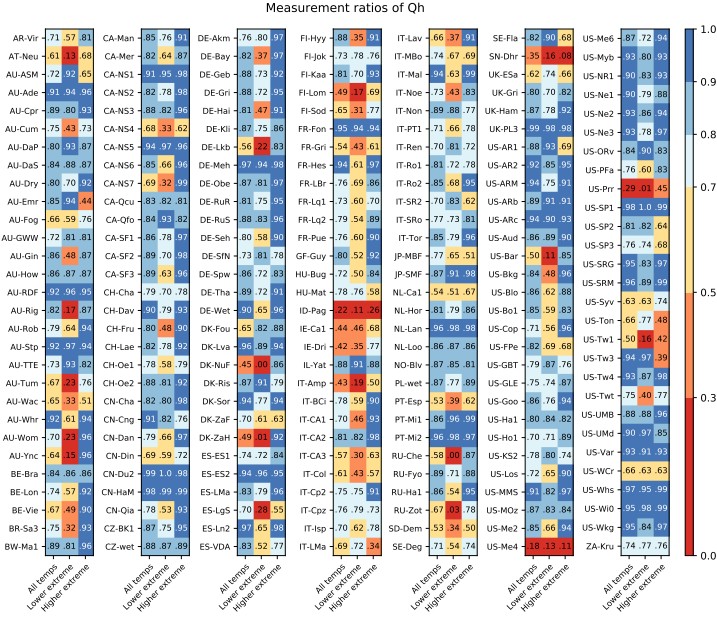

Figure 7: $Q_h$ measurement ratios of flux tower sites for all temperatures, the lower extreme temperatures, and the upper extreme temperatures.





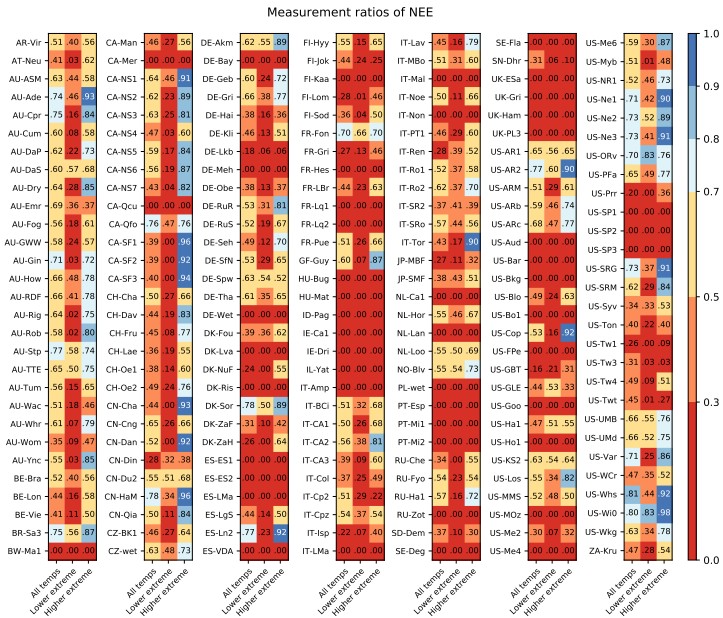

Figure 8: NEE measurement ratios of flux tower sites for all temperatures, the lower extreme temperatures, and the upper extreme temperatures.





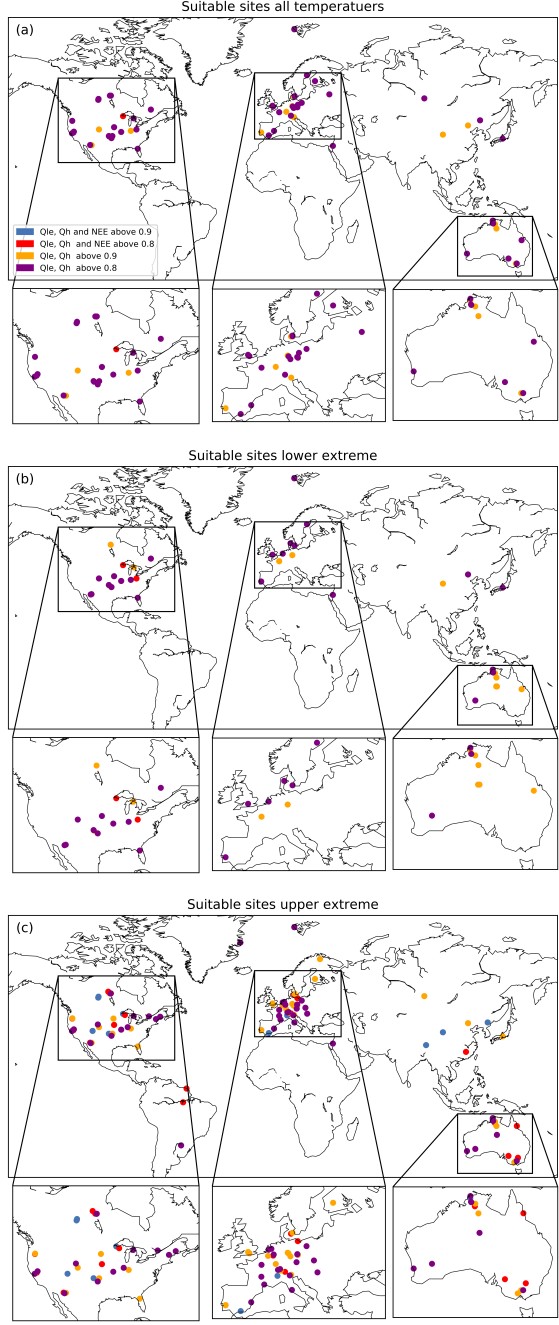

Figure 9: Selection of flux tower with the highest measurement ratios for all temperatures. Sites are selected where $Q_{le}$, $Q_h$, and NEE measurement ratios are all above 0.9 or 0.8, and seperatly where $Q_{le}$ and $Q_h$ are above 0.9 or 0.8. Panel (a) shows sites for all temperatures, panel (b) for lower extreme temperatures and panel (c) for upper extreme temperatures.