# Peer review of "How representative are FLUXNET measurements of surface fluxes during temperature extremes?"

_Biogeosciences, 2018_

## Referee Comment (RC1) · Anonymous Referee #1 · 23 Jan 2019

General comments: The manuscript by van der Horst et al., poses an interesting question about the FLUXNET data and about the representativeness of the flux measurements during temperature extremes. While the topic is of interest in particular to the modelling community, I have a major concern about their approach. The authors explore data availability at each measurement site based on the availability of the temperature, sensible and latent heats, and NEE data. They take the ratio of the available data for heat (latent or sensible) or NEE, relative to the available temperature data, also accessed through FLUXNET, to compare sites. This way data availability is biased by the availability of the temperature data. My questions is why did the authors not use complete temperature records (from meteorological or remote sensing products) for

each site to compare with the absolute availability instead of taking a relative proxy that is a biased by the quality of temperature measurements and is not comparable between sites? The authors themselves suggest this approach to the modelling community in lines 31-33. Specific comments: Page 1 Line 17: Why not using the Tier 2 dataset that is more complete, if this study is focusing on data availability? Page 1 Line 22: Perhaps they mean the "availability" of temperature and not "measurement ratio". Measurement ratio for temperature would be 1 based on their description. Page 3 Lines 5-6: Exactly for this reason, the measurement ratio is relative to each site and cannot be compared across sites. Page 9 line 8: Indeed. But, in my opinion, the authors should have assessed the quality of the flux data independently of the quality of temperature data since the two are measured separately.

---

## Author Comment (AC1) · 29 Jan 2019

We thank Reviewer 1 for their comments and respond below point by point.

General comments: The manuscript by van der Horst et al., poses an interesting question about the FLUXNET data and about the representativeness of the flux measurements during temperature extremes.

Author response: Thank-you!

While the topic is of interest in particular to the modelling community, I have a major concern about their approach. The authors explore data availability at each measurement site based on the availability of the temperature, sensible and latent heats, and NEE data. They take the ratio of the available data for heat (latent or sensible) or NEE, relative to the available temperature data, also accessed through FLUXNET, to compare sites.

Author response: Yes, this is correct: our aim with this study was to independently assess the availability of the measured surface energy and carbon fluxes during extreme conditions.

My questions is why did the authors not use complete temperature records (from meteorological or remote sensing products) for each site to compare with the absolute availability instead of taking a relative proxy that is a biased by the quality of temperature measurements and is not comparable between sites? The authors themselves suggest this approach to the modelling community in lines 31-33.

Author response: There are often many ways to approach a problem and each method has different strengths and weaknesses. In our approach, we were particularly motivated by how these data may be used to develop and evaluate models. Given this motivation, there is no need to generate a complete temperature record, instead our aim was to assess whether during conditions one could run a model (i.e. because we have meteorological forcing data), whether we have matching surface energy and carbon fluxes with which we could evaluate the model output. In revision we will make this case much clearer for the reader.

Author response: The Reviewer was also concerned how differing availability of measured temperatures across sites might bias a site-to-site comparison. This is a fair question although we don't necessarily agree that it affects our motivated comparison (see above). Nevertheless, in revision we will investigate this further and add some appropriate text to the manuscript.

Specific comments: Page 1 Line 17: Why not using the Tier 2 dataset that is more complete, if this study is focusing on data availability?

Author response: We did not use Tier 2 for the simple reason that the Tier 2 data set is not freely available to the community. Those with access to Tier 2 data can use our codes to reproduce our results but we could not use Tier 2 due to the access policies inherent in those data.

Page 1 Line 22: Perhaps they mean the "availability" of temperature and not "measurement ratio". Measurement ratio for temperature would be 1 based on their description.

Author response: We will clarify the associated text in the revised manuscript.

Page 3 Lines 5-6: Exactly for this reason, the measurement ratio is relative to each site and cannot be compared across sites.

Author response: As noted above, we will address this in our revised manuscript.

Page 9 line 8: Indeed. But, in my opinion, the authors should have assessed the quality of the flux data independently of the quality of temperature data since the two are measured separately.

Author response: As stated above, this is perhaps a question of different motivations; our aim was never to assess the temperature record. We acknowledge the reviewer's opinion and we do respect that they may have different motivations here. In revision we will ensure our motivation and framing of the paper is clearer. We will also attempt to address the impact of any biases in temperature record on our results.

---

## Referee Comment (RC2) · Anonymous Referee #2 · 5 Feb 2019

The manuscript by van der Horst et al. addresses the availability of eddy covariance (EC) flux measurements under extreme temperature conditions, whereby 'extreme' is defined relative to each site. The analysis in this manuscript is, in my opinion, very well conducted and the results are described and presented in a very concise and clear manner. Potential caveats and misinterpretation of the results are well explained. This study will certainly be of interest to both the eddy covariance and land surface model community, and it will be very helpful for researchers selecting sites that experience temperature extremes.

I do not have major criticism concerning the overall approach of this study, but some

(mostly minor) suggestions on how the results are presented and discussed:

Second paragraph of the introduction section: I agree that land surface models need a better representation of physiological processes under extreme conditions, but I do not think that eddy covariance data are the key to provide these formulations. In this paragraph it reads a bit as if the authors try to suggest that with the right selection, EC data could be used to parameterize photosynthesis/respiration response curves in land surface model, which would be a huge stretch. EC data provide information on a big-leaf photosynthesis (provided that flux partitioning is correct), which cannot be directly used to parameterize LSMs which require information at leaf level. In addition, I am not sure how well NEE partitioning algorithms are tested/evaluated in extreme conditions. That being said, EC data are of course not useless for land surface modelers, but will be most useful (at least for physiology) in model evaluation/benchmarking, in which emergent canopy-level model results can be directly compared to EC data. The use of these data is mentioned at several points in the manuscript, and I would simply suggest some rewording here to not give the reader a wrong impression of what this study could be used for.

Figure 1: as the authors explain in the results section, the sudden increase in the availability of flux measurements relative to temperature measurements above approx. 40 degrees is mostly caused by a few sites that experience such high temperatures, thus it could be a site selection effect and not a robust pattern. In order to make this clearer in this Figure (and not just in the text), one could present e.g. dashed lines (instead of solid lines), wherever the number of sites/site years is below a certain value, just to give the reader an idea about the robustness of the results at different temperatures.

page 2, l.35: is 'high-frequency' really the right term here?

page 2, l. 40ff: I suggest to list some more relevant studies here. It would be useful to get a better idea of what and how EC measurements taken under extreme conditions

can be used for.

page 3, l. 7ff: '...whether the availability of direct observations...' would be better here?

page 3, l. 33f: I suggest to move this sentence to the last paragraph of the introduction section.

page 4, l. 10: what exactly do you mean by 'lack enough measurements'?

page 5, l. 10f.: so where do the rainfall amounts in Figure 5 come from? Were they calculated from the same measurements that were used in this analysis? Please describe this in more detail here. Please add 'mean annual' precipitation and temperature to axis labels of Figure 5.

page 5, l. 12ff: For me the absence of any relationship is the most striking aspect of Figure 5, and it would make sense to describe this first.

page 5, l. 29ff: presentation of results is very detailed here and could be shortened for the sake of readability.

page 6, l. 4ff: I think these results are also useful for observation-based studies, not only for modeling, which could be mentioned here or later in the paper.

page 7, l. 4: if there is no 'second', remove the 'first'

page 7, l. 27f.: maybe clarify that the removed data points are gapfilled and hence not used for this analysis, which focuses on measured data only.

page 8, l. 6: delete 'are excluded'

page 8, l. 35ff.: this paragraph belongs mostly to the Results section.

Please do not present new results in the Conclusions (l.30ff.)

Figure 9 caption: 'separately', not 'seperatly'

Table S4: Please add whether the data come from the LaThuile or the FLUXNET2015

dataset.

---

## Referee Comment (RC3) · Anonymous Referee #3 · 5 Feb 2019

General comments:

The manuscript by van der Hort et al. gives a very interesting overview of global FLUXNET data availability - within an objectively pre-selected subset of FLUXNET stations - with a focus on how well temperature extremes are represented. The results are of good use for, above all, the modelling community. It might provide help for selecting suitable sites for validating models in the context of climate projections. This is what the authors propose.

I want to highlight that the results are interesting mainly because of their clear yet counter-intuitive character. For instance, the low correlation between the amount of

precipitation as well as mean temperature and the data availability are rather unexpected. Knowing that many factors other than (micro)meteorological play into the overall data availability, I think this study objectively summarizes the results using their own metric for an easy interpretation, without going into too much detail about the reasons (which is not the focus of this manuscript).

As a general comment on the used methodology I want to mention that using temperature measurements as the reference for data availability is confusing at first (as was mentioned by Referee 1). It seemes arbitrary and is clearly biased by the availability of temperature measurements. However, I understand from the author's response that they rely on actually measured data available at the respective FLUXNET site as one possible method. It would help, though, if this is expressed in more detail. After all, temperature is used as 'reference length' of time series, since the metric of measurement ratio requires the temperature to be available. This simple circumstance is not clearly described in the manuscript.

Also, a lot of possible comments related to shortcomings of the study (i.e., the qualitative character of the results) are dispersed with the caveat and comments in anticipation of criticism to certain points in the conclusion.

Minor comments:

p9 l08: It is important to always make clear whether the mean and tails refer to per-site or all sites

p3 l18: 'most observations' should be something like 'longest time series' or similar

p3 l38: A more widely used threshold is 10 Wm2, why is 1 used (instead of straightforwardly zero)? Also, how are erroneous measurements of shortwave radiation at night identified? Was this condition applied only if it seemes erroneous?

p4 l21ff: The numbers given on the following lines referring to Fig. 1, while given as approximates, are still inaccurate. E.g., the maximum ratio for Qh, Qle in 1b is clearly

rather at 30 degC than 20 degC.

p4 l23: 'affect instrumentation' is a very unprecise wording, as the measurements are affected

p4 l26: What is meant by '68 individual measurements'?

p5 l10f: This phrase seems odd and unnecessarily complicated, if meaning not representing the total amount of annual precipitation, please re-phrase.

p5 l14: from 'Qle and Qh, but for NEE, most are in cool sites...' I don't get this phrase. Should it read 'Qle and Qh but for NEE and most are in cool sites...'?

Figure 1: Please show more ticks on the y-axis.
* * *

---

## Author Comment (AC2) · 11 Feb 2019

We think the reviewer for their comments and suggestions. We think there are many useful suggestions in their review - and as the reviewer notes these are largely minor suggestions. We do not see any problems in fully resolving their comments and will do so thoroughly in a revised manuscript.

---

## Author Comment (AC3) · 11 Feb 2019

We thank the reviewer for their comments and efforts in reviewing our manuscript. Virtually all their comments are useful and straightforward to deal with in a revised manuscript. We do however acknowledge the suggestion by the reviewer in responding to Reviewer 1 which we will do. Clearly, the reviewer recognises value in our methods and our metrics but a little additional clarity in the explanation is recognised in responding to reviewer 1. In summary, this is a useful review and we will respond thoroughly in a properly revised manuscript.

---

## Author Response (AR1)

We received three referees' reports. Referees 2 and 3 recommended minor revisions but Referee 1 recommended reject. We address each point made by the referees below.

Our responses are in red, new text added to the manuscript is indented and italicised.

**Anonymous Referee #1**

General comments: The manuscript by van der Horst et al., poses an interesting question about the FLUXNET data and about the representativeness of the flux measurements during temperature extremes. While the topic is of interest in particular to the modelling community, I have a major concern about their approach. The authors explore data availability at each measurement site based on the availability of the temperature, sensible and latent heats, and NEE data. They take the ratio of the available data for heat (latent or sensible) or NEE, relative to the available temperature data, also accessed through FLUXNET, to compare sites.

**Yes, this is correct: our aim with this study was to independently assess the availability of the measured surface energy and carbon fluxes during extreme conditions.**

This way data availability is biased by the availability of the temperature data. My questions is why did the authors not use complete temperature records (from meteorological or remote sensing products) for each site to compare with the absolute availability instead of taking a relative proxy that is a biased by the quality of temperature measurements and is not comparable between sites?

First, we clearly did not express our goals clearly enough and to resolve this we have added several new pieces of text in the introduction to make this clear including:

We use measurements of temperature and  $Q_{le}$ ,  $Q_h$  and NEE from FLUXNET sites because ultimately we seek to identify those sites useful for developing, evaluating and benchmarking LSMs. To do this requires the LSM to be provided meteorological forcing measured concurrently with the fluxes. We therefore cannot blend the measured fluxes with meteorological observations taken elsewhere.

By providing simultaneous and co-located measurements of both the meteorological forcing of the surface, and the associated turbulent energy fluxes, FLUXNET provides a critical resource for the development, evaluation and benchmarking of land surface models.

We use measurements of temperature and  $Q_{le}$ ,  $Q_h$  and NEE from FLUXNET sites because ultimately we seek to identify those sites useful for developing, evaluating and benchmarking LSMs. To do this requires the LSM to be provided meteorological forcing measured concurrently with the fluxes. We therefore cannot blend the measured fluxes with meteorological observations taken elsewhere.

Our goal is to identify those FLUXNET sites with data useful to explore land surface processes under extreme temperature conditions.

We agree that longer records could be achieved by using alternative data sources.

However, weather stations at other locations need not reflect the conditions at the flux tower (and therefore the measured fluxes), whereas remotely sensed products do not directly measure the air temperatures and depending on the product, will have a mismatch in spatial scale compared to the tower footprint ( $\sim$ 1 km2).

Second, we did check for bias and we have added some additional text to note that there is no bias – the temperature data from each FLUXNET site is much more freely available than any of the actual fluxes. We have added two new pieces of text to highlight this.

Overall, temperature observations were available 86% of the time,  $Q_{\text{le},}\,62\%$  of the time,  $Q_h\,68\%$  of the time and NEE 30% of the time.

We examined the availability of measured temperature relative to the potential availability after we excluded sites with less then 8 months of data, less than 50% of data being measured and nighttime. We note 88% of all sites reported measurements for more than 80% of the time. Only 6% of sites had measurements for 50%-70% of the time and we excluded sites with less than 50% from subsequent analysis.

We can display the frequency histogram to demonstrate this – we do not think this should be added to the paper however as the sentences above provide the required information. Clearly, virtually all the sites we use have 80-100% data cover for temperature:

The authors themselves suggest this approach to the modelling community in lines 31-33.

There are often many ways to approach a problem and each method has different strengths and weaknesses. In our approach, we were particularly motivated by how these data may be used to develop, run and evaluate models. Given this motivation,

there is no need to generate a complete temperature record, instead our aim was to assess whether during conditions one could run a model (i.e. because we have meteorological forcing data including observed temperature), whether we have matching surface energy and carbon fluxes with which we could evaluate the model output.

The text the reviewer is highlighting was:

One way forward to resolve how to choose FLUXNET data for extremes is to combine an analysis of meteorological sites with FLUXNET sites. Using sites maintained by meteorological agencies to identify extreme events (e.g. heatwaves) and then interrogate the FLUXNET sites near to the meteorological site for the availability of measurements of  $Q_{le}$ ,  $Q_h$  and NEE could enable a modeller to choose suitable sites for land surface model development and evaluation.

We have added the following text to help clarify the issues here:

While one possible way forward, inconsistencies between observations from meteorological agencies relative to FLUXNET (location, geographical distribution, height of measurements, standardisation of measurements over short grass) highlight the challenges in using meteorological observations that are physically separate from the FLUXNET observations.

Specific comments:

Page 1 Line 17: Why not using the Tier 2 dataset that is more complete, if this study is focusing on data availability?

We did not use Tier 2 for the simple reason that the Tier 2 data set is not freely available to the community. Those with access to Tier 2 data can use our codes to reproduce our results if they wish but we are committed to using freely available data. We have added a statement in the paper to clarify this (Section 2.1):

Only freely-available site datasets from each release were used.

Page 1 Line 22: Perhaps they mean the "availability" of temperature and not "measurement ratio". Measurement ratio for temperature would be 1 based on their description.

No, we meant the measurement ratio, but we have modified this sentence from:

We showed that the measurement ratios for both sensible and latent heat fluxes are generally lower (0.79 and 0.73, respectively) than for temperature measurements, and the ratio of net ecosystem exchange measurements are appreciably lower (0.42)

to

We showed that the measurement ratios for both sensible and latent heat fluxes are generally lower (0.79 and 0.73, respectively) than for temperature

measurements, and the measurement ratio of net ecosystem exchange measurements are appreciably lower (0.42)

(we added the word "measurement" after "temperature")

Page 3 Lines 5-6: Exactly for this reason, the measurement ratio is relative to each site and cannot be compared across sites.

We acknowledge the reviewers statement but we are not clear what the reviewer really means here. The statement we made was:

While eddy-covariance data have been widely used to examine the impact of temperature extremes, the measurement of temperature and the measurement of  $Q_{le}$ ,  $Q_h$  and NEE are independent.

We think that there may be a misunderstanding based on our explanation and to accommodate this we have added the following:

While eddy-covariance data have been widely used to examine the impact of temperature extremes, the measurement of temperature and the measurement of  $Q_{le}$ ,  $Q_h$  and NEE are independent in terms of the instrumentation used. However, the measured temperature is provided in published data, along with measurements for the site of net radiation, wind speed, humidity etc alongside measurements of  $Q_{le}$ ,  $Q_h$  and NEE. A land surface modeller requires all these data to drive a land surface model for evaluation or process-based studies. We are therefore interested in the relationship between measurements of temperature, and in particular extreme temperatures and concurrent measurements of  $Q_{le}$ ,  $Q_h$  and NEE.

Page 9 line 8: Indeed. But, in my opinion, the authors should have assessed the quality of the flux data independently of the quality of temperature data since the two are measured separately.

We accept that this is the reviewer's opinion and we do of course respect that different researchers can approach a question in different ways. From our perspective, while temperature and fluxes are measured by separate instruments, they are not independent of each other in terms of ecosystem behaviour, due to feedbacks between the land and the atmosphere. To build process-based understanding of land surface dynamics during temperature extremes, the co-occurring measurements at flux towers offer a unique opportunity to explore this because the temperature and fluxes are measured at a specific location and then published within a freely accessible data set together. Similarly, a modeler has to use the observed temperature and the observed meteorology together to then use the measured fluxes because the temperature, radiation, wind are all used to force a land model. In the context of how these FLUXNET data are actually used by modellers, using other temperature data would have led to an incompatibility between our results and how the results can actually be used.

We think the reviewer's comment is directed at our statement:

Our analysis poses interesting questions about the FLUXNET data that deserve further exploration.

We think this is a valid statement and it is not clear how we might modify it to reflect the reviewer's opinion given the rest of this paragraph communicates some of these specific questions.

**Anonymous Referee #2**

The manuscript by van der Horst et al. addresses the availability of eddy covariance (EC) flux measurements under extreme temperature conditions, whereby 'extreme' is defined relative to each site. The analysis in this manuscript is, in my opinion, very well conducted and the results are described and presented in a very concise and clear manner. Potential caveats and misinterpretation of the results are well explained. This study will certainly be of interest to both the eddy covariance and land surface model community, and it will be very helpful for researchers selecting sites that experience temperature extremes.

**Thank-you.**

I do not have major criticism concerning the overall approach of this study, but some (mostly minor) suggestions on how the results are presented and discussed:

Second paragraph of the introduction section: I agree that land surface models need a better representation of physiological processes under extreme conditions, but I do not think that eddy covariance data are the key to provide these formulations. In this paragraph it reads a bit as if the authors try to suggest that with the right selection, EC data could be used to parameterize photosynthesis/respiration response curves in land surface model, which would be a huge stretch. EC data provide information on a bigleaf photosynthesis (provided that flux partitioning is correct), which cannot be directly used to parameterize LSMs which require information at leaf level. In addition, I am not sure how well NEE partitioning algorithms are tested/evaluated in extreme conditions. That being said, EC data are of course not useless for land surface modelers, but will be most useful (at least for physiology) in model evaluation/benchmarking, in which emergent canopy-level model results can be directly compared to EC data. The use of these data is mentioned at several points in the manuscript, and I would simply suggest some rewording here to not give the reader a wrong impression of what this study could be used for.

**We had to choose to either delete the text or improve it a little to address this reviewer's comment and we concluded on balance to add a little further detail. We added:**

Although model improvements in the representation of physiological responses to temperatures are likely to be driven by data from leaf-level and manipulation experiments, data from eddy-covariance is also of value. For example, Keenan et al. (2019) recently quantified an apparent inhibition of daytime ecosystem respiration, showing that the diurnal pattern differed from expectations using the global FLUXNET network.

Figure 1: as the authors explain in the results section, the sudden increase in the availability of flux measurements relative to temperature measurements above approx. 40 degrees is mostly caused by a few sites that experience such high temperatures, thus it could be a site selection effect and not a robust pattern. In order to make this clearer in this Figure (and not just in the text), one could present e.g. dashed lines (instead of solid lines), wherever the number of sites/site years is below a certain value, just to give the reader an idea about the robustness of the results at

different temperatures.

We have taken this advice on board and have revised the figure to use dashed lines where the sample size is less than 1000. The choice of 1000 is somewhat arbitrary of course.

page 2, I.35: is 'high-frequency' really the right term here?

We have modified this text to avoid the term while retaining the important text:

From each FLUXNET site, measurements of the exchange of latent heat flux  $(Q_{le})$ , sensible heat flux  $(Q_h)$  and net ecosystem exchange (NEE) are available at 30- to 60-minute resolution, alongside meteorological variables (including air temperature, net radiation, precipitation and relative humidity).

page 2, I. 40ff: I suggest to list some more relevant studies here. It would be useful to get a better idea of what and how EC measurements taken under extreme conditions can be used for.

We have taken this advice on board and added further examples.

page 3, I. 7ff: '...whether the availability of direct observations...' would be better here?

We modified this section to accommodate Reviewer 1's criticisms and so we have not made this edit.

page 3, I. 33f: I suggest to move this sentence to the last paragraph of the introduction section.

We agree that we should move this sentence – but feel it is better at the beginning of the last paragraph of the introduction, rather than at the very end.

page 4, I. 10: what exactly do you mean by 'lack enough measurements'?

We mean there are sometimes very few observations to base analyses on if we use a 2 standard deviation threshold.

We have modified the text to read:

We did repeat our analysis using exactly the two standard deviations; this led to some qualitative differences in our results because some sites lack enough measurements to provide reliable results where the temperature distribution was not normally distributed.

page 5, I. 10f.: so where do the rainfall amounts in Figure 5 come from? Were they calculated from the same measurements that were used in this analysis? Please describe this in more detail here. Please add 'mean annual' precipitation and temperature to axis labels of Figure 5.

The data were sourced from FLUXNET and are from the same measurements as used in the other analysis. We have added appropriate text. We have also amended the axes as suggested.

page 5, I. 12ff: For me the absence of any relationship is the most striking aspect of

Figure 5, and it would make sense to describe this first.

We have restructured this paragraph to follow the advice of the reviewer and now talk about the absence of any relationship first.

page 5, I. 29ff: presentation of results is very detailed here and could be shortened for the sake of readability.

We agree it is quite detailed – it is actually a lot shorter than we originally drafted to try to help readability. We have not further reduced this part of the text in the revised version as on balance we are keen to provide the detail, but we are open to further advice from the editor.

page 6, I. 4ff: I think these results are also useful for observation-based studies, not only for modeling, which could be mentioned here or later in the paper.

Yes – we fully agree and it was not our intention to suggest otherwise. We have edited this sentence on the basis that we to touch on the value of our results for observational programs later. Specifically, at the end of the discussion, but we are also reluctant to be seen to "direct" the observational community, so we are attempting to retain balance.

The edit introduced is minor – we have simply edited this sentence from:

The FLUXNET eddy covariance flux measurements are among the most valuable observations available for developing, evaluating and benchmarking land surface models.

to

The FLUXNET eddy covariance flux measurements are among the most valuable observations available for understanding processes, and for developing, evaluating and benchmarking land surface models.

We hope this suffices but if we have misunderstood the reviewer we are happy to take further advice on board.

page 7, I. 4: if there is no 'second', remove the 'first'

We have made this correction

page 7, l. 27f.: maybe clarify that the removed data points are gap filled and hence not used for this analysis, which focuses on measured data only.

Yes – we have followed the reviewer's advice and made this clear by adding a sentence at the end of the paragraph:

We avoid the consequences of these procedures in quality controlling and gap-filling data by only using those data that are directly observed.

page 8, I. 6: delete 'are excluded'

Yes – thanks, we have corrected this.

page 8, I. 35ff.: this paragraph belongs mostly to the Results section.

We have followed the reviewer's advice and moved this paragraph to the results section.

Please do not present new results in the Conclusions (I.30ff.)

These are not results from our paper, they are contextualizing our results and so would not be appropriate in the results section. We have now moved this to the discussion section.

Figure 9 caption: 'separately', not 'separatly'

Thank-you – corrected.

Table S4: Please add whether the data come from the LaThuile or the FLUXNET2015 data set

Yes – we have added a column with this information to Table S4.

**Anonymous Referee #3**

General comments:

The manuscript by van der Hort et al. gives a very interesting overview of global FLUXNET data availability - within an objectively pre-selected subset of FLUXNET stations - with a focus on how well temperature extremes are represented. The results are of good use for, above all, the modelling community. It might provide help for selecting suitable sites for validating models in the context of climate projections. This is what the authors propose.

**Thanks for these positive comments**

I want to highlight that the results are interesting mainly because of their clear yet counter-intuitive character. For instance, the low correlation between the amount of precipitation as well as mean temperature and the data availability are rather unexpected.

**Yes – and in responding to Reviewer 2 some changes in the text's order has helped highlight this.**

Knowing that many factors other than (micro)meteorological play into the overall data availability, I think this study objectively summarizes the results using their own metric for an easy interpretation, without going into too much detail about the reasons (which is not the focus of this manuscript).

**Yes – thank-you. That was our intent.**

As a general comment on the used methodology I want to mention that using temperature measurements as the reference for data availability is confusing at first (as was mentioned by Referee 1). It seems arbitrary and is clearly biased by the availability of temperature measurements. However, I understand from the author's response that they rely on actually measured data available at the respective FLUXNET site as one possible method. It would help, though, if this is expressed in more detail. After all, temperature is used as 'reference length' of time series, since the metric of measurement ratio requires the temperature to be available. This simple

circumstance is not clearly described in the manuscript.

These comments relate to the comments by Reviewer 1. We have responded in detail to Reviewer 1 and hope that these changes will satisfy reviewer 3 too.

Also, a lot of possible comments related to shortcomings of the study (i.e., the qualitative character of the results) are dispersed with the caveat and comments in anticipation of criticism to certain points in the conclusion.

Yes – we obviously agree with the reviewer here.

Minor comments:

p9 l08: It is important to always make clear whether the mean and tails refer to persite or all sites

Yes - we have followed the reviewer's advice and added clarification.

p3 I18: 'most observations' should be something like 'longest time series' or similar

Yes – "most" is too vague. We have modified the last two sentences to read:

We therefore seek to identify which FLUXNET site data are most suitable for analysing processes under extreme temperature conditions with the goal of identifying those sites most useful for land surface model development and evaluation of the surface energy, water and carbon budgets during extreme temperatures.

p3 l38: A more widely used threshold is 10 Wm2, why is 1 used (instead of straightforwardly zero)? Also, how are erroneous measurements of shortwave radiation at night identified? Was this condition applied only if it seems erroneous?

**Our text reads:**

Given our focus on  $Q_{le}$ ,  $Q_h$  and NEE we excluded night-time data where incoming shortwave radiation was < 1 W m-2. In case of erroneous measurements of shortwave radiation at night we excluded all data between 11 pm and 6 am.

We did not use 10 W m-2 for identifying night time because this seems a very high flux to use to exclude night time. That said, we see that our text was not clear and we have modified it to:

Given our focus on  $Q_{le}$ ,  $Q_h$  and NEE we excluded night-time data using two criteria. We first excluded all data between 11 pm and 6 am local time. In addition, if shortwave radiation was < 1 W m-2 for an individual time period then associated measurements were also excluded. This did not exclude many measurements as shortwave radiation was rarely reported as non-zero at night but there were occasional shortwave radiation > 1 W m-2 in observations at night.

p4 I21ff: The numbers given on the following lines referring to Fig. 1, while given as approximates, are still inaccurate. E.g., the maximum ratio for Qh, Qle in 1b is clearly rather at 30 degC than 20 degC.

We agree the precision on our approximations could have been better and we have modified these in the text. However, the maximum ratio for Qh and Qle in Figure 1b is at  $20^{\circ}$ C – and this then remains very similar through to about  $30^{\circ}$ C. The modified text now reads:

At the lowest temperatures, ratios for  $Q_{le}$  and  $Q_h$  range from ~0 to ~0.3 but these increase as temperatures increase to maximum ratios of ~0.8 at around  $20^{\circ}$ C and remain at ~0.8 through to  $30^{\circ}$ C. For NEE ratios increase to ~0.6 at around  $35^{\circ}$ C for NEE. A minor dip in  $Q_{le}$  and  $Q_h$  ratios occurs at  $0^{\circ}$ C associated with the phase change of water, which most likely affects the operation of instrumentation. At the upper extreme of the temperature distribution, ratios decline between  $30^{\circ}$ C and  $45^{\circ}$ C from ~0.8 to ~0.6 for  $Q_{le}$ and  $Q_h$  and from ~0.6 to ~0.5 for NEE.

p4 I23: 'affect instrumentation' is a very unprecise wording, as the measurements are affected

Yes - we have modified this to

"which most likely affects the operation of instrumentation"

p4 I26: What is meant by '68 individual measurements'?

We mean, there are 68 actual reported measurements at this site at a temperature in excess of the 44°C threshold. We have revised the sentence to make this clear.

This peak is associated with temperatures > 44°C, which are rare and associated with measurements at Au-Cpr (there are only 68 individual measurements in excess of 44°C at this site), AU-GWW (23 individual measurements), AU-Stp (24 individual measurements) and SN-Dhr (33 individual measurements).

p5 I10f: This phrase seems odd and unnecessarily complicated, if meaning not representing the total amount of annual precipitation, please re-phrase.

We agree and have re-written the sentence:

Note the amounts of rainfall shown in Figure 5 are accumulated only over times when temperature data are selected and therefore cannot be compared with observations taken at meteorological stations.

p5 I14: from 'Qle and Qh, but for NEE, most are in cool sites...' I don't get this phrase. Should it read 'Qle and Qh but for NEE and most are in cool sites...'?

We agree that this was unclear. In part to help simplify our results section we decided this was a sentence that could be deleted.

Figure 1: Please show more ticks on the y-axis.

We have followed the reviewer's suggestion.

**How representative are FLUXNET measurements of surface fluxes during temperature extremes?**

Sophie V.J. van der Horst1, Andrew J. Pitman2,3, Martin G. De Kauwe2,3, Anna Ukkola2,4, Gab Abramowitz2,3, Peter Isaac4

1Meteorology and Air Quality, Wageningen University, 6700 HB, Wageningen, the Netherlands.
 2ARC Centre of Excellence for Climate Extremes
 3Climate Change Research Centre, University of New South Wales, Sydney, NSW, 2052, Australia
 4Research School of Earth Sciences, Australian National University, Canberra, ACT, 2601 Australia

4OzFlux Central Node, TERN Ecosystem Processes, Melbourne, VIC 3159, Australia

10

5

Correspondence to: Andrew J. Pitman (a.pitman@unsw.edu.au)

Abstract. In response to a warming climate, temperature extremes are changing in many regions of the world. Therefore,

- 15 understanding how the fluxes of sensible heat, latent heat and net ecosystem exchange respond and contribute to these changes is important. We examined 216 sites from the open access Tier 1 FLUXNET2015 and Free-Fair-Use La Thuile datasets, focussing only on observed (non-gap filled) data periods. We examined the availability of sensible heat, latent heat and net ecosystem exchange observations coincident in time with measured temperature for all temperatures, and separately for the upper and lower tail of the temperature distribution and expressed this availability as a measurement ratio. We
- 20 showed that the measurement ratios for both sensible and latent heat fluxes are generally lower (0.79 and 0.73 respectively) than for temperature measurements, and the measurement ratio of net ecosystem exchange measurements are appreciably lower (0.42). However, sites do exist with a high proportion of measured sensible and latent heat fluxes, mostly over the United States, Europe and Australia. Few sites have a high proportion of measured fluxes at the lower tail of the temperature distribution over very cold regions (e.g. Alaska, Russia) or at the upper tail in many warm regions (e.g. Central America and
- 25 the majority of the Mediterranean region), and many of the world's coldest and hottest regions are not represented in the freely available FLUXNET data at all (e.g. India, the Gulf States, Greenland and Antarctica). However, some sites do provide measured fluxes at extreme temperatures suggesting an opportunity for the FLUXNET community to share strategies to increase measurement availability at the tails of the temperature distribution. We also highlight a wide discrepancy between the measurement ratios across FLUXNET sites that is not related to the actual temperature or rainfall

1

30 regimes at the site, which we cannot explain. Our analysis provides guidance to help select eddy covariance sites for researchers interested in understanding and/or modelling responses to temperature extremes. Martin De Kauwe 21/2/2019 12:28 PM Deleted: 3

Martin De Kauwe 21/2/2019 12:27 PM Deleted: and Martin De Kauwe 21/2/2019 12:27 PM Deleted: 3ARC Centre of Excellence for Climate

Extremes, and

Andy Pitman 26/2/2019 9:39 AM

Andy Pitman 26/2/2019 9:39 AM Deleted: exploring

**1** Introduction**

Changes in the upper and lower tails of the temperature distribution are key characteristics of how global warming will impact climate (Hartmann et al. 2013). These expected changes in temperature are in line with a series of recent high-profile extremes witnessed across Europe (2003, 2010; Coumou and Rahmstorf, 2012; Schär et al. 2004; Barriopedro et al. 2011),

- 5 western North America (van Mantgem et al. 2009), the Amazon (2005, 2010; Philips et al. 2009; Lewis et al. 2011) and Australia (2012/2013; van Gorsel et al. 2018). Changes in temperature extremes are not only limited to the warm tail; the cold tail has also seen a notable change, with observed decreases in cold extremes particularly across North America (Wolter et al. 2015). Given the wide-ranging impacts of temperature on vegetation function (Berry and Björkman, 1980; Gunderson et al., 2009; Valladares et al., 2014; Van Gorsel et al., 2016; Kumarathunge et al., in press), health (McMichael and
- 10 Lindgren, 2011), socio-economics (McEvoy et al., 2012; Colombo et al., 1999; Zander et al., 2015) and land-atmosphere feedbacks (Fischer et al., 2007; Teuling et al., 2010; Miralles et al., 2012; Kala et al., 2016; Donat et al. 2017), projecting the impact of changes in temperature extremes is critical.

Our understanding of how temperature extremes will change is based on simulations using coupled climate models, e.g. the Coupled Model Intercomparison Project (CMIP5) (Eyring et al., 2016). To build confidence in these projections, models should be consistent with our understanding of changing temperature extremes, the impact on the vegetation and the associated feedback on the climate. However, current models are known to have key weaknesses in simulating both temperature extremes (Sillmann et al., 2013; Sippel et al., 2017) and the response of the vegetation to these extremes. For example, most climate models represent broad geographic regions with a single photosynthetic temperature response

- 20 function, which varies only with plant functional type (Smith and Dukes 2013; Lombardozzi et al. 2015; Mercado et al. 2018). This assumption would seemingly contradict empirical evidence showing that the temperature response of photosynthesis varies as a function of climate (Berry & Björkman, 1980; Gunderson *et al.*, 2009). Furthermore, studies show that plants adjust their temperature response of photosynthesis and respiration to changes in ambient temperature (Way and Sage 2008; Lombardozzi et al. 2015). Although model improvements in the representation of physiological responses to
- 25 temperatures are likely to be driven by data from leaf-level and manipulation experiments, data from eddy-covariance is also of value. For example, Keenan et al. (2019) recently quantified an apparent inhibition of daytime ecosystem respiration, showing that the diurnal pattern differed from expectations using the global FLUXNET network.

Improving how well models simulate temperature extremes, and how vegetation responds to these extremes requires
empirical data. The global network of eddy-covariance towers (commonly known as FLUXNET), which includes over 900 sites and over 7000 site years (fluxdata.org, 2018) provides measurements of the exchange of carbon, energy and water between the land and the atmosphere. Therefore, eddy covariance measurements provide our best ecosystem-scale estimate of the vegetation's response to heat extremes (Ciais et al. 2005; Teuling et al. 2010; Wolf et al. 2013; von Buttlar et al. 2018; Flach et al. 2018; De Kauwe et al. 2019) although some limitations inevitably remain (*e.g.* lack of energy closure, see

[revised manuscript text omitted]

**Andy Pitman 26/2/2019 10:12 AM Moved (insertion) [2]**

Andy Pitman 26/2/2019 10:12 AM Deleted: thisFor example, the land surface is important in simulating Andy Pitman 26/2/2019 10:12 AM Moved up [2]: Ukkola et al. (2016) used FLUXNET data to show systematic errors in how well models captured this

Andy Pitman 26/2/2019 10:12 AM

Andy Pitman 26/2/2019 10:12 AM **Deleted:** transition of the vegetation during periods of water stress.

**Andy Pitman 20/2/2019 12:42 PM **Deleted:** In this study, we focus on**

Deleted. In this study, we locus on

**Andy Pitman 27/2/2019 2:35 PM Deleted:**

Anna Ukkola 26/2/2019 3:02 PM Deleted: ultimately we seek to identify those sites useful for developing, evaluating and benchmarking LSMs. To do this requires the LSM to be provided meteorological forcing measured concurrently with the fluxes

Anna Ukkola 26/2/2019 3:02 PM Deleted: c Anna Ukkola 26/2/2019 3:02 PM Deleted: an

Andy Pitman 27/2/2019 2:35 PM **Deleted:** Andy Pitman 20/2/2019 1:50 PM

Andy Pitman 20/2/2019 1:50 PM

Andy Pitman 21/2/2019 10:41 AM

Andy Pitman 21/2/2019 10:39 AM

Andy Pitman 21/2/2019 10:39 AM

Andy Pitman 21/2/2019 10:40 AM

Andy Pitman 21/2/2019 10:41 AM **Deleted:** elements of

[revised manuscript text omitted]

40 data are selected and therefore cannot be compared with observations taken at meteorological stations. Figure 5 shows little relationship between temperature or rainfall and the measurement ratios. For example, some cool dry sites have high measurement ratios whereas others have low ratios. Similarly, some hot wet sites have high and some have low ratios for both Qle and Qh and for the upper tail of NEE. Few sites have high ratios for the overall temperature distribution or for the Andy Pitman 21/2/2019 11:05 AM Deleted: , Andy Pitman 21/2/2019 11:05 AM Deleted: Q Andy Pitman 21/2/2019 11:05 AM Deleted: and NEE Andy Pitman 21/2/2019 11:05 AM Deleted: for Qk and Qh Andy Pitman 21/2/2019 11:05 AM Deleted: and Andy Pitman 21/2/2019 11:06 AM Deleted: 0 Andy Pitman 21/2/2019 11:06 AM Deleted: 0

Andy Pitman 21/2/2019 11:15 AM **Deleted:** Since only a sub-set of Andy Pitman 21/2/2019 11:15 AM **Deleted:** 
[revised manuscript text omitted]

20

---

## Author Response (AR2)

Thanks for the additional comments by the reviewer. We are obviously very happy with the advice from the reviewer.

They had two comments. The text change recommended has been made and how reads:

Although model improvements in the representation of physiological responses to temperatures need to be informed by data from leaf-level and manipulation experiments, data from eddy-covariance is also of value.

The second comment relates to the apparent discontinuity in Figure 1b. There actually is no issue here, although we do understand why the reviewer asked the question. Figure 1b is reproduced below – on the left as in the manuscript and on the right with a solid line to demonstrate no discontinuity exists where the dashed line becomes a solid line in the left hand panel.